# Differences in the Effects of Calcium and Magnesium Ions on the Anammox Granular Properties to Alleviate Salinity Stress

Yeonju Kim [1], Jaecheul Yu [2,3], Soyeon Jeong [2], Jeongmi Kim [2], Seongjae Park [2], Hyokwan Bae [2],
Sung-Keun Rhee [4], Tatsuya Unno [5], Shou-Qing Ni [6] and Taeho Lee [2,*]

1   Disaster Scientific Investigation Division, National Disaster Management Research Institute,
    Ulsan 44538, Korea; kyj9219@gmail.com
2   Department of Civil and Environmental Engineering, Pusan National University, Busan 46241, Korea;
    yjcall0715@pusan.ac.kr (J.Y.); jeongsy002@gmail.com (S.J.); magicstar2@nate.com (J.K.);
    sjker25@naver.com (S.P.); hyokwan.bae@pusan.ac.kr (H.B.)
3   Institute for Environment and Energy, Pusan National University, Busan 46241, Korea
4   Department of Biological Sciences and Biotechnology, Chungbuk National University, Cheongju 28644, Korea;
    rhees@chungbuk.ac.kr
5   Faculty of Biotechnology, School of Life Sciences, SARI, Jeju National University, Jeju City 63243, Korea;
    tatsu@jejunu.ac.kr
6   School of Environmental Science and Engineering, Shandong University, Jinan 250012, China;
    sqni@sdu.edu.cn
*   Correspondence: leeth55@pusan.ac.kr; Tel.: +82-51-510-2465

**Abstract:** Divalent cations were known to alleviate salinity stress on anammox bacteria. Understanding the mechanism of reducing the salinity stress on anammox granules is essential for the application of the anammox process for saline wastewater treatment. In this study, the effect of $Ca^{2+}$ and $Mg^{2+}$ augmentation on the recovery of the activity of freshwater anammox granules affected by salinity stress was evaluated. At the condition of a salinity stress of 5 g NaCl/L, the specific anammox activity (SAA) of the granule decreased to 50% of that of the SAA without NaCl treatment. Augmentation of $Ca^{2+}$ at the optimum concentration of 200 mg/L increased the SAA up to 78% of the original activity, while the augmentation of $Mg^{2+}$ at the optimum concentration of 70 mg/L increased the SAA up to 71%. EPS production in the granules was increased by the augmentation of divalent cations compared with the granules affected by salinity stress. In the soluble EPS, the ratio of protein to polysaccharides was higher in the granules augmented by $Ca^{2+}$ than with $Mg^{2+}$, and the functional groups of the EPS differed from each other. The amount of $Na^+$ sequestered in the soluble EPS was increased by the augmentation of divalent cations, which seems to contribute to the alleviation of salinity stress. Ca. *Kuenenia*-like anammox bacteria, which were known to be salinity stress-tolerant, were predominant in the granules and there was no significant difference in the microbial community of the granules by the salinity stress treatment. Our results suggest that the alleviation effect of the divalent cations on the salinity stress on the anammox granules might be associated with the increased production of different EPS rather than in changes to the anammox bacteria.

**Keywords:** anammox; cation; EPS; granule; SAA; salinity stress

## 1. Introduction

Saline wastewater from various industries such as food processing, petroleum, chemical, and pharmaceuticals accounts for approximately 5% of industrial wastewater [1–3]. In particular, the discharge of untreated high-salinity wastewater including high nitrogen concentrations can deteriorate the water quality and cause eutrophication in rivers and coasts [3,4]. Recently, the anaerobic ammonium oxidation process (Anammox), which is a cost-effective nitrogen removal technology, has attracted considerable attention for the treatment of saline wastewater containing a high concentration of nitrogen [1].

Anammox bacteria can be applied to saline wastewater treatment, but they have a lower salt resistance than other bacteria. In general, the anammox bacteria can be divided into marine and freshwater species. Although the marine species have a strong tolerance to high salinity, their growth rate is slower than twice that of the freshwater species, so there is a limit to nitrogen-rich wastewater treatment [5]. On the other hand, freshwater species with relatively fast growth rates are used widely for nitrogen-rich wastewater treatment. Hence, step-by-step salt adaptation and salt stress alleviation methods are required for nitrogen-rich saline wastewater treatment, owing to their low salinity tolerance.

Compatible solutes (CS), hydroxylamine, and cations have often been applied to alleviate the salinity stress on fresh anammox bacteria (FAB) [6]. The addition of CS, such as glycine betaine, trehalose, and ectoine, increases the osmotic potential of anammox bacteria cells to protect them from the saline osmotic pressure. Despite this, CS and organic matter can promote the growth of heterotrophic denitrifying bacteria and cause nitrite competition with the anammox bacteria. The recovery time of the anammox reactor shocked with salt could be shortened by 50% after adding hydroxylamine [6]. Nevertheless, this may be a temporary effect according to the supply of the substrate because the anammox bacteria convert hydroxylamine, an intermediate product of the anammox reaction, to ammonium and nitrogen gas. In addition, it is difficult to expect a longterm stable effect of the supply of hydroxylamine for a stable anammox reaction in a saline environment.

The addition of cations, such as $K^+$, $Fe^{2+}$, $Ca^{2+}$, and $Mg^{2+}$, can alleviate the salinity stress of the anammox granules exposed to the salt environment. $K^+$ is often applied to improve biological wastewater processes under a saline environment. On the other hand, under the individual shock of NaCl and KCl for FAB, the $IC_{50}$ (the salt concentration that caused the 50% inhibition on anammox activity) for KCl (0.096 M) was lower than that for NaCl (0.106 M) [7,8], suggesting that FAB may be affected more by $K^+$ than $Na^+$ under a salt environment where both ions exist. $Fe^{2+}$ also promotes the anammox process performance by inducing heme C and hydrazine dehydrogenase production by the anammox bacteria [9]. Although $Ca^{2+}$ and $Mg^{2+}$ were reported to accelerate the aggregation of anaerobic microorganisms and promote the formation of granules [10,11], their effective concentrations varied depending on the studies. Hence, it is unclear how they contribute to the alleviation of the salinity stress of the anammox granules in a saline environment.

This study examined the effects of $Ca^{2+}$ and $Mg^{2+}$ additions on FAB activity under a saline environment. The mechanism of both cation-induced salinity stress alleviations in the anammox process was probed by examining the granule properties and the microbial community.

## 2. Materials and Methods

### 2.1. Anammox Granules

Anammox granules of the laboratory-scale anammox reactor that has been operating stably with synthetic wastewater (influent 200 mg $NH_4^+$-N/L and 200 mg $NO_2^-$-N/L) for more than one year were used in this study. This parent reactor was operated in a constant temperature room maintained at $33 \pm 1$ °C. An average nitrogen removal efficiency and nitrogen removal rate of 85% and 1.4 kg N/m$^3$/day, respectively, were achieved.

### 2.2. Batch Test

A batch test was performed in a 125 mL vial (working volume 100 mL) and 3 g/L biomass from the parent reactor was inoculated. The test was conducted in three phases to evaluate and understand the effects of cations on the salt stress of the anammox granules in a saline environment. First, the activity of the anammox granules were analyzed according to the salt concentrations (1, 3, 5, 10, 15, and 20 g NaCl/L) to determine the salinity stress level for the anammox granules (phase 1). At the salt concentrations corresponding to the $IC_{50}$, the activity of the anammox granules according to $Ca^{2+}$ (30, 70, 100, 200, and 300 mg/L) and $Mg^{2+}$ (20, 50, 70, and 100 mg/L) was evaluated (phase 2). Then, the effects of cations on the morphology, extracellular polymeric substances (EPS) content

and composition, and microbial community structure of the anammox granules were investigated at the optimum cation concentration (C3 for 200 mg $Ca^{2+}$/L and 5 g NaCl/L and C4 for 70 mg $Mg^{2+}$/L and 5 g NaCl/L) (phase 3). C1 without salt and C2 with only salt (5 g NaCl/L) were used as controls.

The batch test was performed for 1 h in phase 1 and for 6 h in phases 2 and 3. All experiments were performed in a constant temperature room maintained at $35 \pm 1$ °C. The composition of the medium was as follows: 100 mg/L $NH_4^+$-N, 100 mg/L $NO_2^-$-N, 1.010 g/L $NaHCO_3$, 0.055 g/L $KH_2PO_4$, 0.005 g/L $CaCl_2 \cdot 5H_2O$, 0.5 g/L, $MgSO_4 \cdot 7H_2O$, 0.010 g/L $FeSO_4 \cdot 7H_2O$, and 0.005 g/L EDTA.

### 2.3. Specific ANAMMOX Activity (SAA)

The specific anammox activity (*SAA*) was analyzed using a respirometer (BRS-100, EETech Co., Chuncheon, Korea). $N_2$ gas production was calculated using the actual gas ($N_2$) concentration measured using a respirometer according to the user's manual (Equation (1)).

$$N_2(molN_2/min) = \frac{P \cdot \Delta N}{R \cdot T \cdot \Delta t} \tag{1}$$

where $\Delta N$ is the net increase in $N_2$ gas volume (L) measured from a respirometer over the reaction time; $\Delta t$ is reaction time (min); $R$ is the ideal gas coefficient (0.082 atm·L/mol·K); $T$ is the temperature (K); $P$ is the pressure (1 atm).

The *SAA* was calculated from the $N_2$ gas production rate divided by the biomass concentrations (gVSS/L) in the vial (Equation (2) [12] and the relative *SAA* (*rSAA*, %) was calculated by Equation (3).

$$SAA \ (gN_2/gVSS/day) = \frac{N_2 \cdot M \cdot f}{XV_L} \tag{2}$$

$$rSAA(\%) = \left(1 - \frac{SAA_c}{SAA_0}\right) \times 100 \tag{3}$$

where $X$ is the biomass concentration (gVSS/L); $M$ is the molecular weight of nitrogen gas (28 g/mol); $f$ is the time modifying factor (1440 min/day); $V_L$ is the volume of liquid (0.1 L); $SAA_0$ is the *SAA* when the salt concentration is zero; $SAA_c$ is the *SAA* at each concentration. All experiments were conducted in duplicate.

### 2.4. Analysis

Biomass concentration (gVSS/L) was analyzed using to the standard method (APHA, 2005). All liquid samples from each vial were filtered through 0.22 μm disposable filters (RephIQuik syringe filter, RephiLe Biosience. Ltd., Shanghai, China) and stored in micro-centrifuge tubes before further analysis. The ammonium and nitrite concentrations were analyzed using a nitrogen analysis kit (Humas Co. Ltd., Daejeon, Korea) according to the standard method [13].

The morphology of the anammox granules under different conditions was observed by scanning electron microscopy (NeoScope Benchtop SEM, JCM-7000, Peabody, MA, USA). The EPS of the granules were extracted using the heating method [14]. The polysaccharide (PS) content in the EPS was analyzed using the phenol sulfuric acid method and the protein (PN) content was analyzed with a modified Lowry colorimetric method. Both the PS and PN contents were estimated using a UV/Vis spectrophotometer (HS-3300, Humas Co. Ltd., Daejeon, Korea). Finally, the total EPS obtained the sum of the PS and PN. The functional structure of the EPS was estimated by the attenuated total reflection (ATR) using an infrared spectrophotometer (Nicolet iS50, Thermo, Waltham, MA, USA). The metal ion contents within the EPS of the samples were determined by inductively coupled plasma optical emission spectrometry (ICP-OES, Optima 8330, PerkinElmer, Waltham, MA, USA) [15].

Statistical analysis was performed using R [16]. The significance of the differences in the EPS according to the addition of salt and cations was evaluated by comparing ANOVA test *p*-values ($p$-value $< 0.05$).

### 2.5. Microbial Community Analysis

The anammox granules were collected at the end of the ANAMMOX activity test. The DNA of all samples was extracted using DNeasy PowerSoil Kit (DNeasy Kits, DEV) on a clean bench to prevent contamination from outside at room temperature. The extracted DNA concentration and purity of the extracted DNA were obtained from the absorbance using a NanoDrop spectrometer (ND-1000, Thermo Fisher Scientific, Waltham, MA, USA). The A260/A280 ratio was a good value between 1.8 and 2.1.

PCR for identifying the microbial community was carried out using amplified 16S rRNA by requesting a combination of the primer set Amx338F (5′-ACT CCT ACG GGC AG3′) and Amx806R (5′-GAC TAC HCH CGT WCT AT-3′). The PCR conditions were as follows: 95 °C for 3 min, followed by 30 cycles of denaturation at 95 °C for 30 s, 55 °C annealing for 30 s and 72 °C for 45s extension, followed by a final extension at 72 °C for 10 min. The samples were implemented in accordance with the Illumina 16S rRNA sequencing library protocol (Macrogen Co., Seoul, Korea). Input gDNA is amplified with 16S Amx338F-Amx806R primers and a subsequent limited-cycle amplification step is performed to add multiplexing indices and Illumina sequencing adapters. The final products are normalized and pooled using the PicoGreen and the size of libraries are verified using the TapeStation DNA screentape D1000 (Agilent). The samples were then sequenced using the MiSeq™ platform (Illumina, San Diego, CA, USA). After the sequencing is completed, MiSeq raw data is classified by sample using an index sequence. Sequences less than 400 bp in length were removed. The obtained sequence is CD-HIT-EST based on the operational taxonomic units ($OTU_s$) analysis program CD-HIT-OUT. Low quality sequences and ambiguous sequences were considered as sequencing errors. After removing the chimera sequence, and the like, clustering sequences having more than 97% sequence similarity to the species level OTU was formed.

### 3. Results and Discussions

### 3.1. Inhibition of Specific Anammox Activity by Salinity Stress

The SAA decreased with increasing salt concentrations (Figure 1a). Up to 10 g NaCl/L, the SAA decreased sharply by more than 60%, and then gradually by approximately 80% at 20 g NaCl/L. In general, the high salinity increases the osmotic pressure. This can inhibit the enzyme activity, and eventually the cell collapses in on itself, which can decrease the bacterial activity and lead to its death [7,17]. The activity of the FAB used in this study also decreased, possibly due to an increase in the osmotic pressure with the increase of the NaCl concentration.

Although it is difficult to compare the $IC_{50}$ values between this and previous studies, since there were different inocula and operating conditions, various $IC_{50}$ values have been reported, such as 3.8 g NaCl/L [18], 5.4 g NaCl/L [19], and 13.5 g NaCl/L [12]. In this study, the $IC_{50}$ was calculated to be approximately 4.8 g NaCl/L (Figure 1), which was used as the salinity stress condition for the anammox granules, and further evaluation of the augmentation of the divalent cations on the SAA was conducted at this concentration.

### 3.2. Effect of Divalent Cations on Alleviation of Salinity Stress

The augmentation of divalent cations ($Ca^{2+}$ and $Mg^{2+}$) relieved the salinity stress of the anammox caused by the $IC_{50}$ (5 g NaCl/L) but the mitigation effect differed according to the ion (Figure 2). The rSAA by the augmentation of $Ca^{2+}$ increased to 77.9% (0.105 $gN_2$/gVSS/day for the SAA at 200 mg $Ca^{2+}$/L) with increasing the $Ca^{2+}$ from 30 to 200 mg/L but decreased again to 66.4% (0.084 $gN_2$/gVSS/day for the SAA) at 300 mg $Ca^{2+}$/L (Figure 2a). In the case of $Mg^{2+}$ augmentation, the rSAA increased slightly to 71% (0.084 $gN_2$/gVSS/day for the SAA at 70 mg $Mg^{2+}$/L) with increasing the $Mg^{2+}$ from

20 mg/L to 70 mg/L but decreased to 55% (0.067 gN$_2$/gVSS/day for the SAA) at 100 mg Mg$^{2+}$/L. It is known that Ca$^{2+}$ above an appropriate level inhibits the energy and the substrate metabolism of anammox bacteria [20], and Mg$^{2+}$ addition causes the breakdown of the granules [21]. However, in this study, the saline stress alleviation effect by the Ca$^{2+}$ addition was higher than that by Mg$^{2+}$. Moreover, the optimal Ca$^{2+}$ and Mg$^{2+}$ concentrations for alleviating salinity stress were determined to be 200 mg/L and 70 mg/L, respectively.

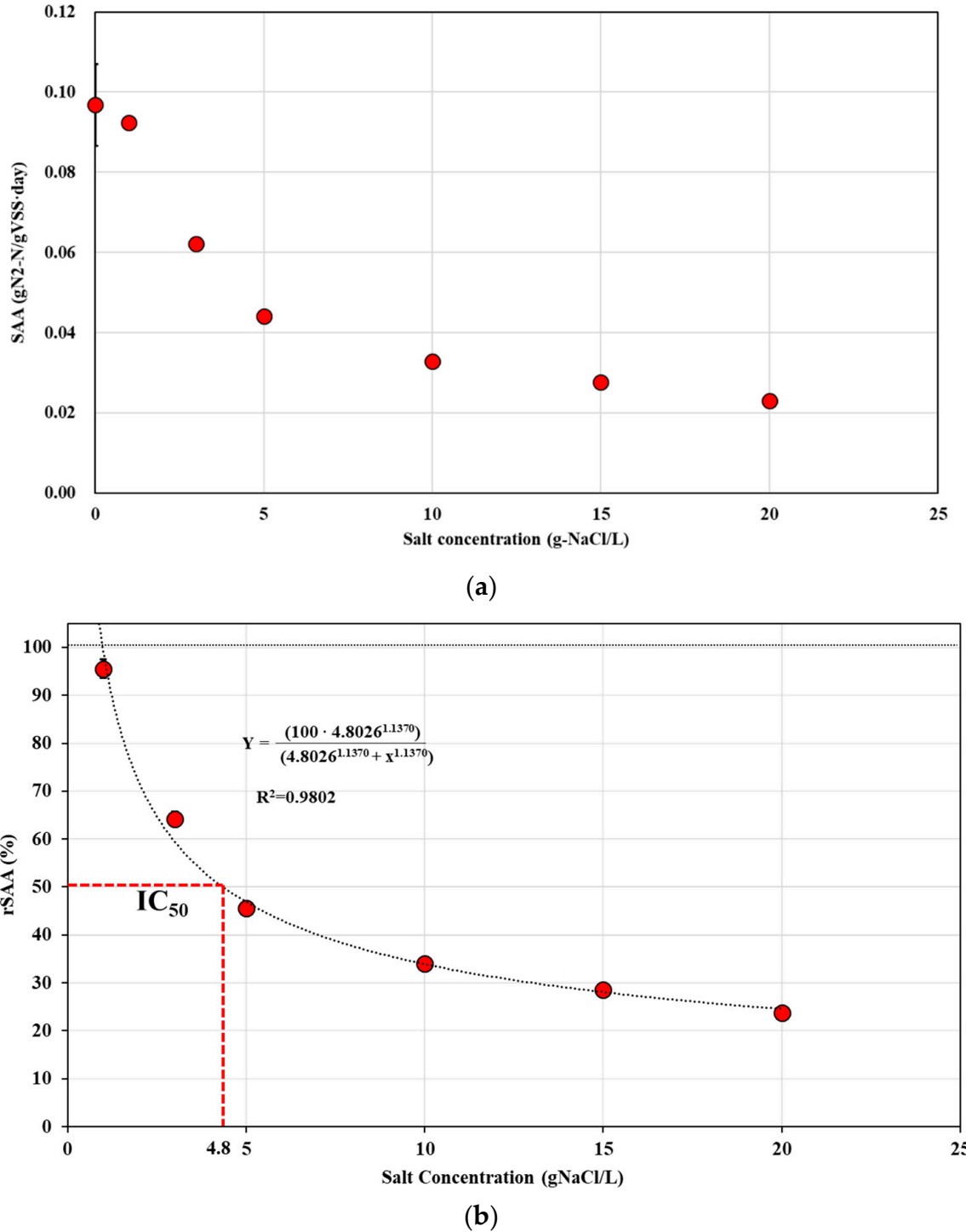

**Figure 1.** Effect of the salt concentration (g NaCl/L) on the specific anammox activity (rSAA, gN$_2$-N/gVSS-day) (**a**) and relative specific anammox activity (rSAA, %) (**b**).

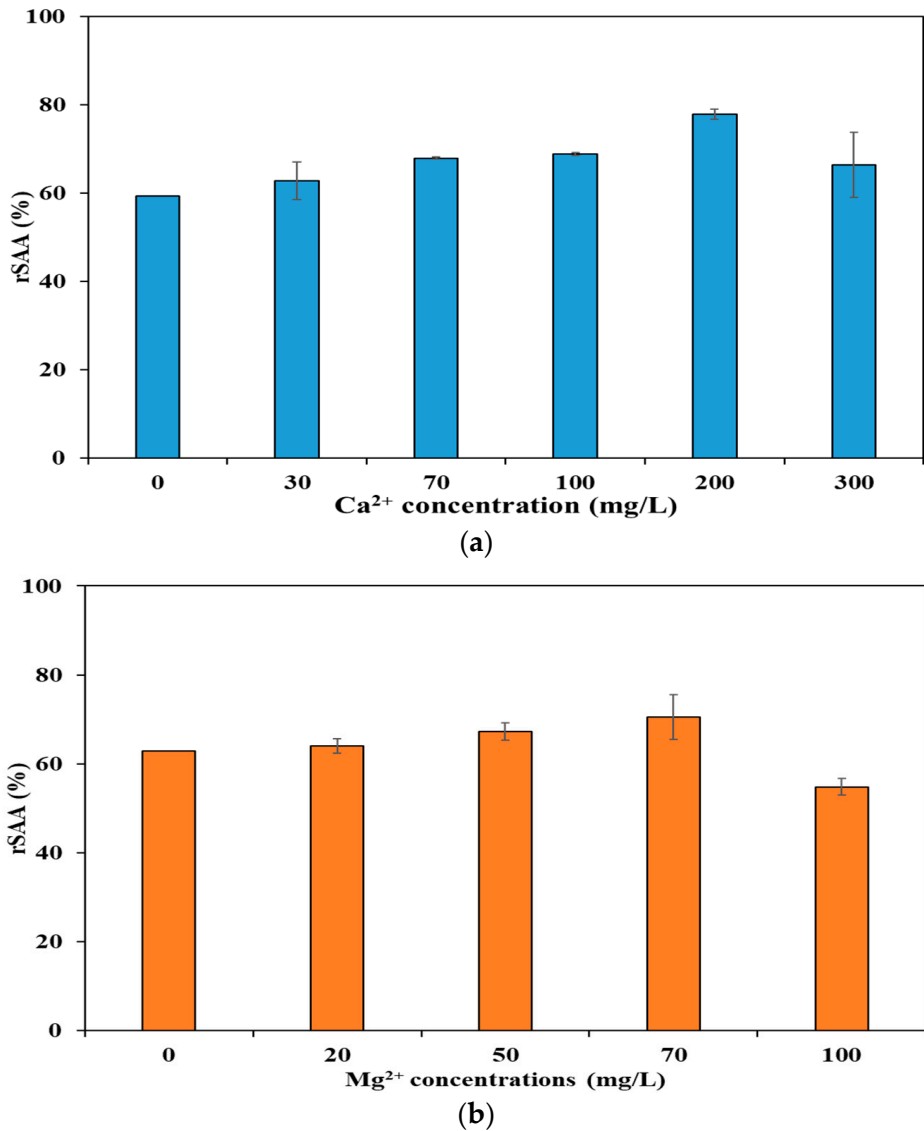

**Figure 2.** Effect of the $Ca^{2+}$ (**a**) and $Mg^{2+}$ (**b**) concentrations on relative specific anammox activity (rSAA, %) at 5 g/L of salt concentration.

The SEM showed that the salinity stress shrank the anammox granule size and roughed its surface (compare Figure 3a,b). When the cations were added, fewer changes in the granule shape and surface evenness were observed (Figure 3c,d). When the divalent cations were added, the granule size and the surface properties became similar with that of the control without the NaCl treatment (Figure 3e,f). Although both the divalent cations contributed to the alleviation of the salinity stress, a denser granular structure was observed in the granules augmented by $Ca^{2+}$ than by $Mg^{2+}$. In previous studies, the addition of $Ca^{2+}$ (100 mg/L) promoted the secretion of the EPS from the activated sludge [22], and the sludge granulation was enhanced by $Ca^{2+}$ (>150 mg/L) [23]. In addition, $Ca^{2+}$ was observed to play a role in a cross-bridge connecting bacteria and the EPS to protect the aggregates of the granules [20]. It is widely known that EPS plays an important role in granulation and substrate transfer, as well as acts as the first barrier against adverse stress [24,25]. $Mg^{2+}$ was known to be essential for phosphodiesterase synthesis to produce bis-(3′-5′)-cyclic dimeric guanosine monophosphate, which acted as an important regulator for the anammox activity under unfavorable conditions [26–28].

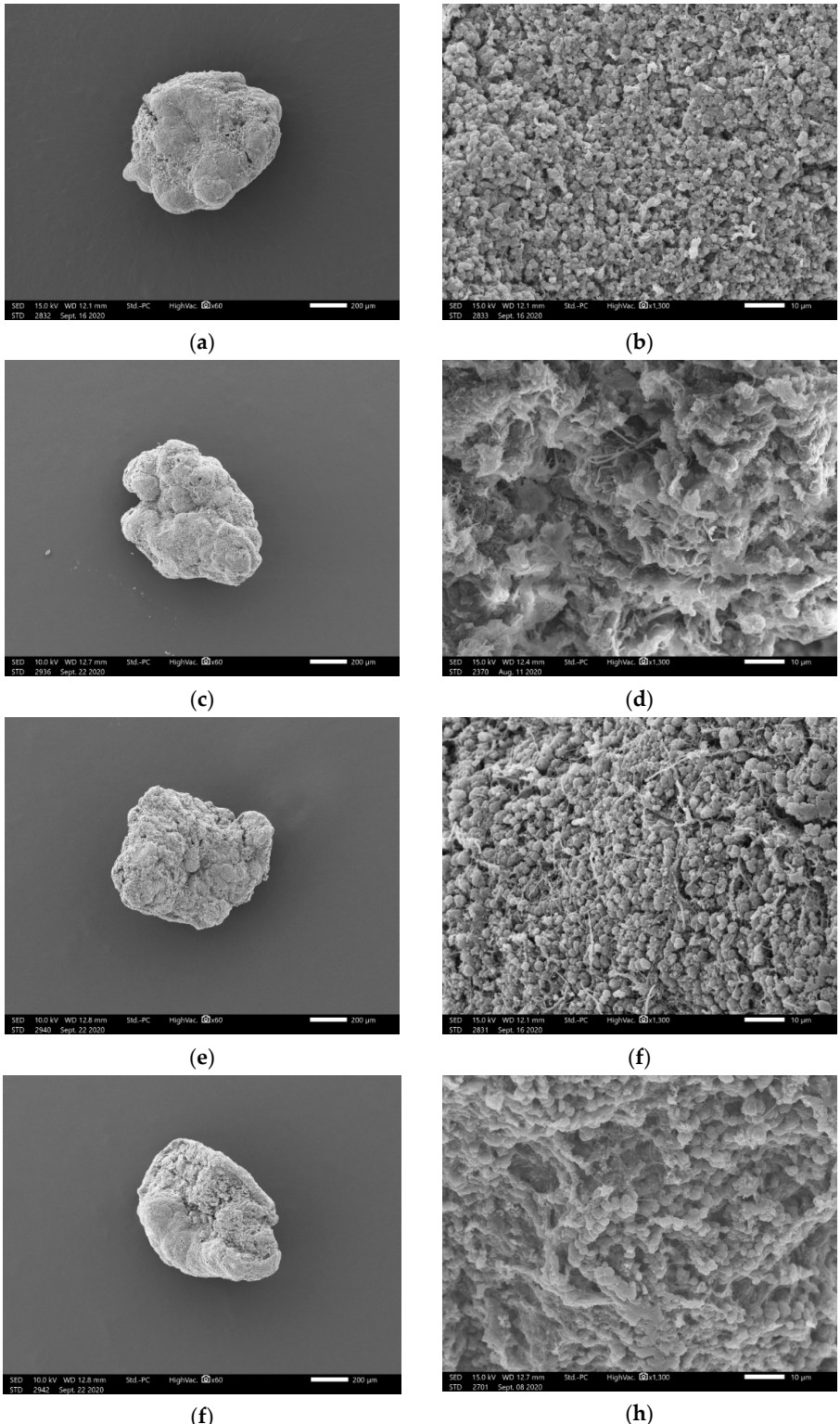

**Figure 3.** SEM images of particles and surface for anammox granules according to cation additions at 5 g NaCl/L: (**a**) The particles of an anammox granule without salt; (**b**) The surface of an anammox granule without salt; (**c**) The particles of an anammox granule at 5 g NaCl; (**d**) The surface of an anammox granule at 5 g NaCl/L; (**e**) The particles of an anammox granule with 200 mg/L $Ca^{2+}$ at 5 g NaCl/L; (**f**) The surface of an anammox granule with 200 mg/L $Ca^{2+}$ at 5 g NaCl; (**g**) The particles of an anammox granule with 70 mg/L $Mg^{2+}$ at 5 g NaCl/L; (**h**) The surface of an anammox granule with 70 mg/L $Mg^{2+}$ at 5 g NaCl/L.

### 3.3. Effect of Divalent Cations on EPS Production and Compositions

The EPS can be generally divided into the tightly bound EPS (B-EPS) and the outer-layer soluble EPS (S-EPS) [29]. In this study, the EPS amount and content of the anammox granules were varied by the salinity stress and following the augmentation of divalent cations (Figure 4). Under salinity stress conditions (5 g NaCl/L), the anammox granules (C2) produced more EPS than the normal granules (C1) under the non-salinity stress conditions. The amount of the EPS from the granules under the salinity stress was increased further by adding 200 mg/L of $Ca^{2+}$ (C3) and was also increased by 70 mg/L of $Mg^{2+}$ (C4). This suggests that effect of divalent cations might be associated with changes in the amount and composition of the EPS. $Mg^{2+}$ augmentation increased the production of the EPS more than $Ca^{2+}$ augmentation for both the B-EPS and S-EPS. In the anammox granules under all conditions, the amount of the B-EPS was approximately 8–10 times higher than those of the S-EPS. Previous studies have shown similar trends. The amount of the B-EPS was approximately 4–20 times higher than those of the S-EPS in the anammox UASB reactors under saline conditions (2.5–15 mg NaCl/L) with $Ca^{2+}$ [30].

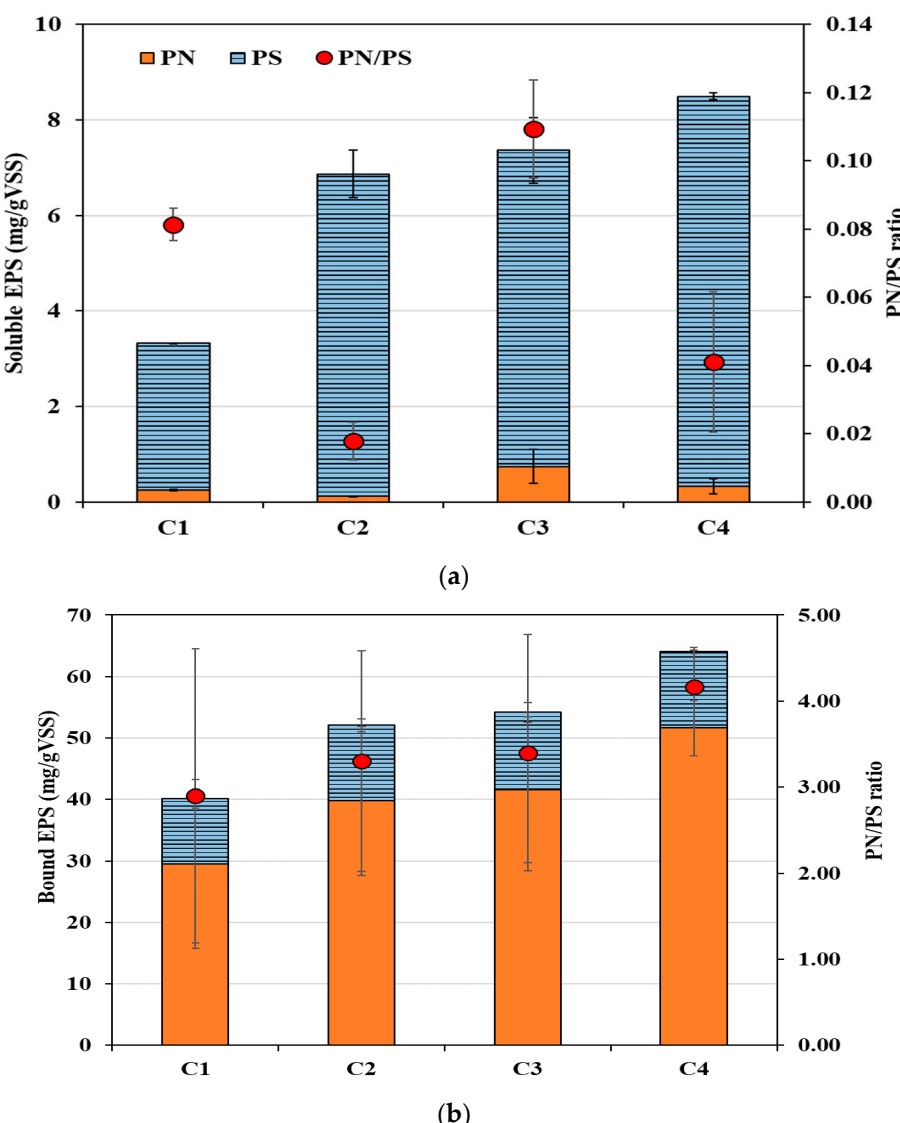

**Figure 4.** PS and PN concentrations, and PN/PS ratio in soluble EPS (**a**) and bound EPS (**b**) anammox granules at C1 (no salt and no cation), C2 (5 g NaCl/L and no cation), C3 (5 g NaCl/L and 200 mg/L $Ca^{2+}$), and C4 (5 g NaCl/L and 70 mg/L $Mg^{2+}$).

Both EPSs are composed mainly of PN and PS [31]. Usually, the PS are reported to contribute to the hydrophilicity of the EPS, whereas the PN is the hydrophobic component [25,32,33]. The decrease in the PN/PS ratio in the EPS led to poor cell surface hydrophobicity, thereby decreasing the flocculating ability of the granule. The PN/PS ratios in the B-EPS and S-EPS in this study were varied depending on the augmented divalent cations. Those changes in the EPS affect the properties and the aggregation of the anammox granules based on the variation in the PN/PS ratios, which may vary depending on the environmental conditions [34,35]. In all conditions, the PN/PS ratio of the B-EPS was relatively high, but it was difficult to find a correlation with the SAA. However, in the case of the S-EPS, although the amount of the S-EPS was small, C1 and C3 showed a relatively high PN/PS ratio, which showed the same trend as the variations of the SAA at each condition. As a result of the ANOVA test, it was confirmed that there was a statistically significant difference because the *p*-value (0.0086) for the PN/PS ratio of the S-EPS was less than 0.05. Principally, there is a statistically significant difference because the *p*-values for the PN/PS ratio between C1 and C2 (*p*-value = 0.0277), C2 and C3 (*p*-value = 0.0082), and C3 and C4 (*p*-value = 0.03072) were less than 0.05. However, the p-value (0.0086) for the PN/PS ratio of the B-EPS was greater than 0.05, indicating that there was no statistically significant difference. $Ca^{2+}$ and $Mg^{2+}$ would have a difference in the salinity stress alleviation because $Ca^{2+}$ and $Mg^{2+}$ showed different PN and PS functional groups in the S-EPS. This suggests that the S-EPS plays a more important role in maintaining anammox activity than the B-EPS under saline conditions.

In addition, the EPS in microbial aggregates have many functional groups (carboxyl, proteins, carbohydrates, and others). FTIR spectra were analyzed to investigate the detailed variations in the PN and PS of the EPS. The functional groups of the EPS were varied by the condition of the salinity stress and following the divalent cation augmentation (Figure 5). In the case of the B-EPS, the anammox granules under the non-salinity stress (C1) and the granules under the salinity stress augmented by divalent cations (C3 and C4) showed similar patterns of FTIR spectra, while the granules under the salinity stress (C2) showed differences in FTIR spectra. The amide group with C-O (1454 $cm^{-1}$), the carboxyl group (C=O, 1403 $cm^{-1}$), and the polysaccharide group (1040 $cm^{-1}$) were observed at C2 (Figure 5a). The S-EPS showed various patterns of FTIR spectra depending on the different salinity stress conditions (Figure 5b). In particular, the amide group with C-O (1454 $cm^{-1}$) disappeared at C2 and was only observed at C3. The polysaccharide group (1040 $cm^{-1}$) was only observed at C2.

Although the peak of the FTIR spectra of the previous studies were different because the operating conditions and the seeding sources were not the same, several studies have shown the variation of the functional groups in the EPS under a saline environment. Polysaccharides and carboxyl groups are known to influence bacterial aggregation through cationic bridging interactions, and protein groups are known to influence the hydrophobic properties of cells required for granule aggregation [36,37]. Although it is difficult to know exactly how the EPS functional group contributes to the alleviation of salt inhibition only with FTIR results, it can be confirmed that the change in the functional group according to the cation addition contributed to alleviate salt stress.

The amount of $Na^+$ associated with EPS increased sharply under the salinity stress conditions, and it was further increased following the augmentation of divalent cations (Figure 6). Previous studies have reported a similar finding of $Na^+$ concentration increases in the EPS of granules when exposing the granules under a salt environment, which significantly inhibited the binding between the multivalent metal ions and the EPS [35]. The amounts of $Na^+$ associated with the S-EPS were approximately three times higher than those of the B-EPS. The amounts of $Na^+$ associated with the B-EPS increased from 5 mg/L at C1 to 73 mg/L at C2 and increased to approximately 100 mg/L at C3 and C4 (after divalent cations augmentation). On the other hand, the amounts of $Na^+$ associated with the S-EPS increased significantly from 46 mg/L at C1 to 240 mg/L at C2 and increased further to 285 mg/L at C3 and C4. Although the reason for why the amounts of $Na^+$ associated

with EPS increased when divalent cations were augmented to granules with salinity stress is unclear, the salinity stress was alleviated by the sequestration of Na$^+$ in the EPS and by preventing it from diffusing into the cell. Consequently, the S-EPS appears to contribute more to the alleviation of salinity stress on anammox granules than the B-EPS.

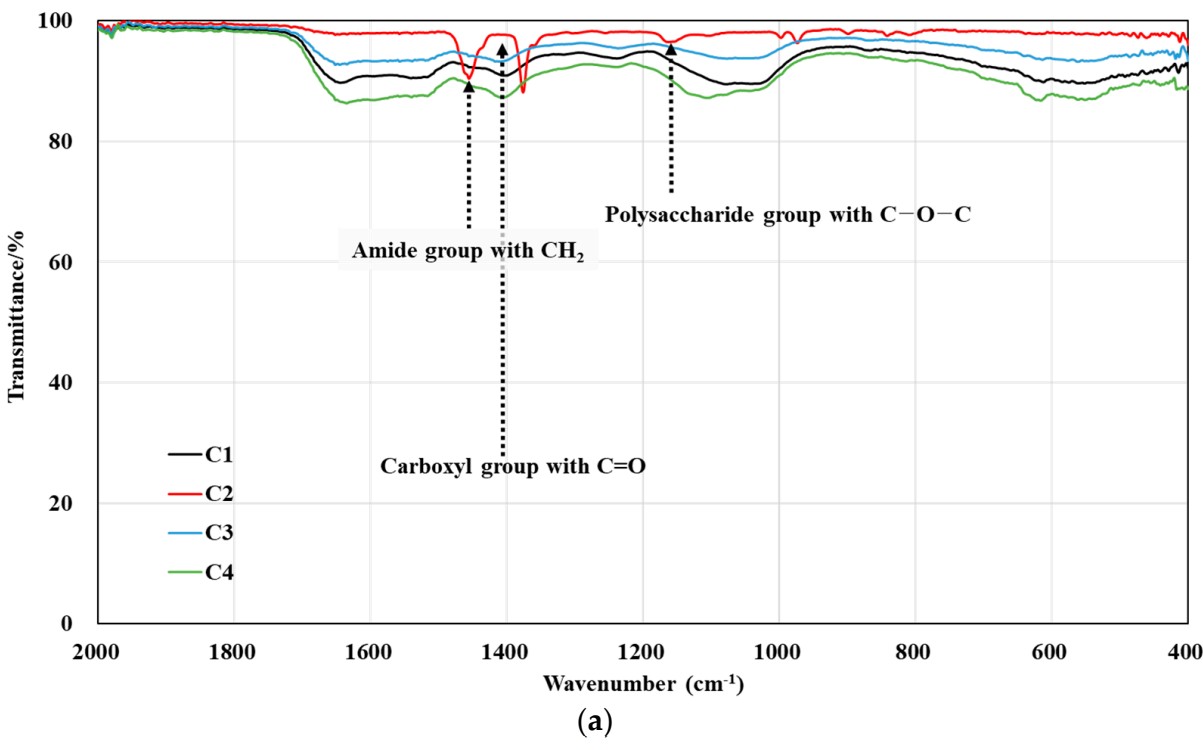

**(a)**

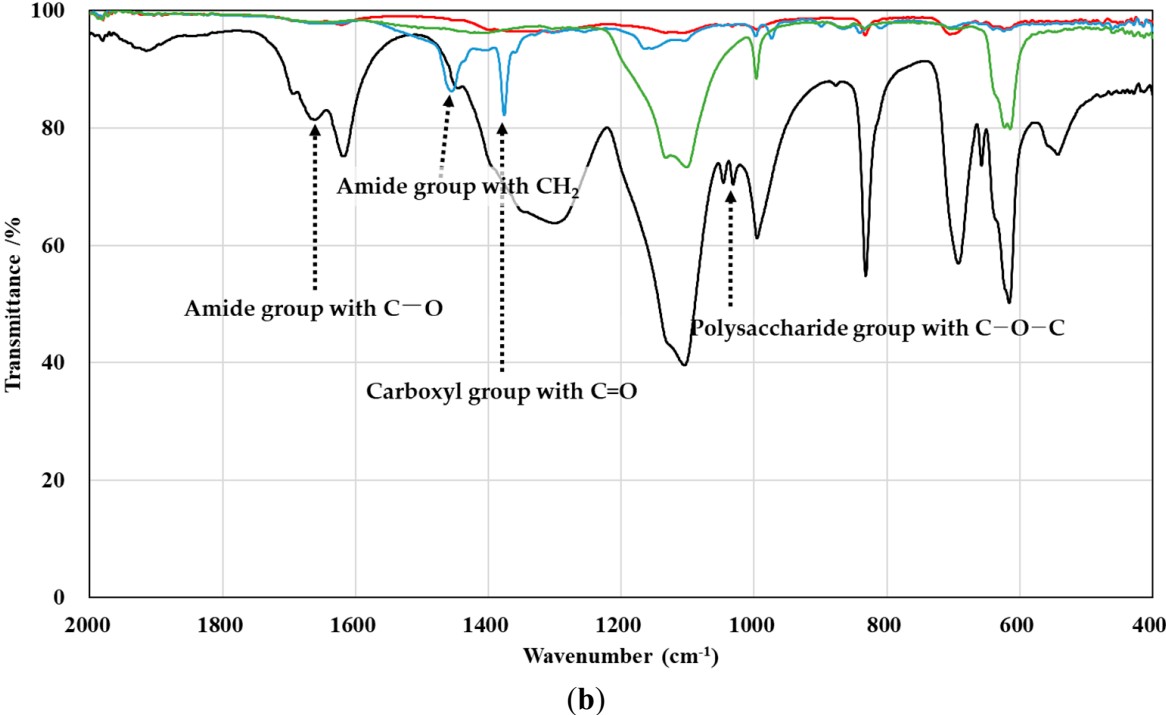

**(b)**

**Figure 5.** FTIR spectra of bound EPS (**a**) and soluble EPS (**b**) at C1 (no salt and no cation), C2 (5 g NaCl/L and no cation), C3 (5 g NaCl/L and 200 mg/L Ca$^{2+}$), and C4 (5 g NaCl/L and 70 mg/L Mg$^{2+}$).

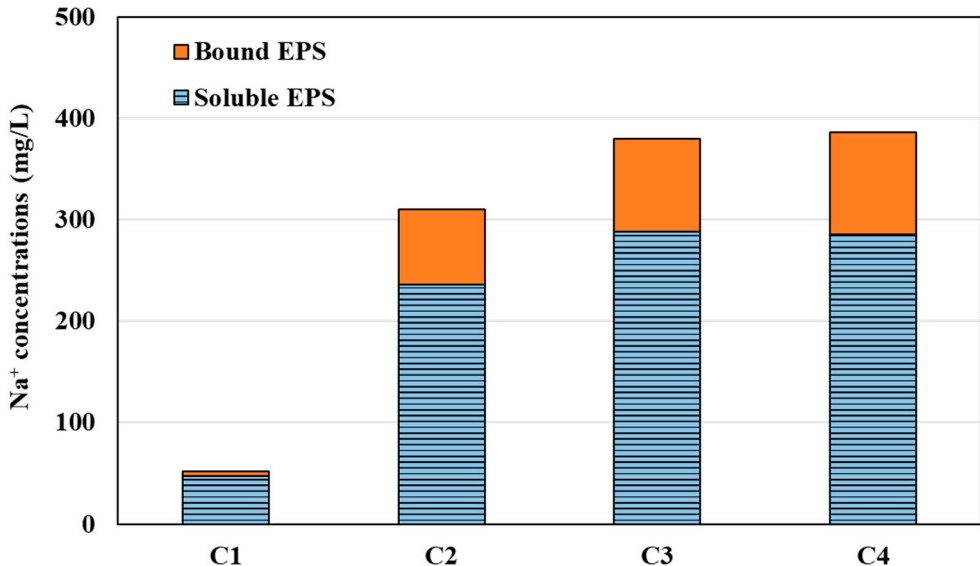

**Figure 6.** Na⁺ concentration in bound EPS and soluble EPS at C1 (no salt and no cation), C2 (5 g NaCl/L and no cation), C3 (5 g NaCl/L and 200 mg/L $Ca^{2+}$), and C4 (5 g NaCl/L and 70 mg/L $Mg^{2+}$).

### 3.4. Effect of Divalent Cations on Microbial Community Compositions

The microbial community composition of the anammox granules was relatively unchanged at either the phylum or species levels, regardless of the experimental conditions (Figure 7). *Planctomycetes*-harboring anammox bacteria accounted for the largest distribution (51.1–56.0%) in the anammox granules, followed by *Chloroflexi* (23.7–29.3%) and *Ignavibacteriae* (7.2~9.2%). The remaining phyla were present at less than 3%. In general, the phyla *Chloroflexi* and *Ignavibacteriae* were frequently observed as major members in the anammox bioreactors [38]. *Chloroflexi* and *Ignavibacteriae* might reduce nitrate to nitrite, which can be used by the anammox bacteria [38]. In addition, *Chloroflexi*, typical filamentous bacterium, might provide a stabilizing backbone, cores, or carriers for the microbial aggregates, and use the soluble microbial products and the EPS released from the decaying anammox bacteria as a carbon source [39–41]. In this study, because *Chloroflexi* was the second largest member in our anammox granules, they would play an important role in maintaining the stability of the anammox granules.

No significant differences were observed in the community composition of the anammox bacteria in the anammox granules (Figure 7b). More than 97% of the phylum *Planctomycetes* were found to be anammox bacteria. Uncultured anaerobic ammonium-oxidizing bacterium (accession no. LC192412) related to Ca. *Kuenenia stuttgartiensis* showed the largest component (68.5–71.6%), followed by Ca. *Brocadia* (16.1–19.1%), Ca. *Jettenia* (3.0–6.7%), and Ca. *Kuenenia* (4.6–6.9%). It is known that Ca. *Kuenenia* can grow well at 15–20 g NaCl/L and can be acclimated at 30 g NaCl/L [42,43]. Ca. *Jettenia* is sensitive even at 4 g NaCl/L [44]. Ca. *Brocadia* has a relatively low salinity stress tolerance [8].

As the salt concentration was increased from 0 g NaCl/L to 30 g/NaCl, Ca. *Kuenenia* increased from 0.02% to 36.9% while Ca. *Jettenia* only increased to 0.27% and Ca. *Brocadia* decreased from 1.92% to 0.04% [45]. In this study, however, there was no significant change in the community composition of the anammox bacteria in our experimental conditions. The experimental time was too short to observe the variations of the microbial (anammox bacteria) community composition. These results suggest that the alleviation of salinity stress by the augmentation of divalent cations might be not caused by microbial community changes.

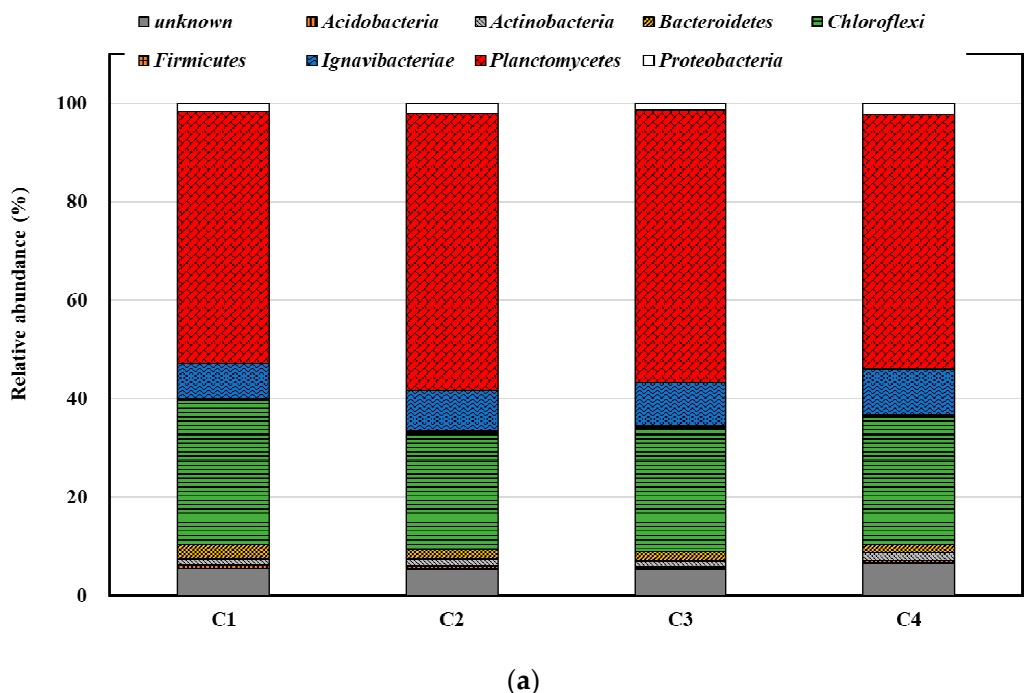

(a)

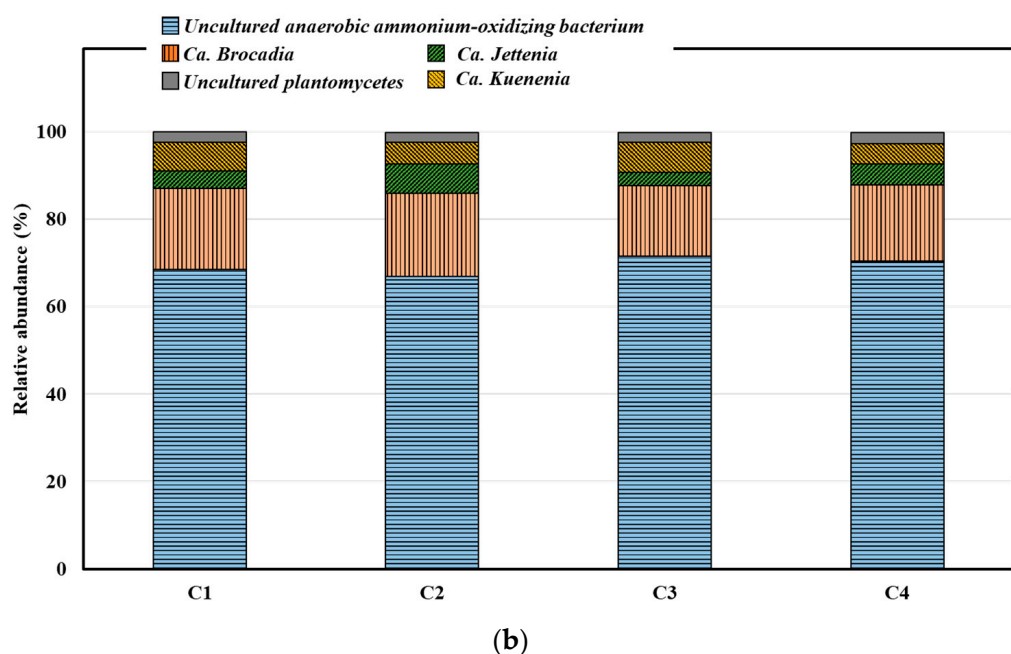

(b)

**Figure 7.** Relative abundances (%) at phylum level (**a**) and at species level on anammox bacteria (**b**); C1 (no salt and no cation), C2 (5 g NaCl/L and no cation), C3 (5 g NaCl/L and 200 mg/L $Ca^{2+}$), and C4 (5 g NaCl/L and 70 mg/L $Mg^{2+}$).

## 4. Conclusions

The augmentation of $Ca^{2+}$ and $Mg^{2+}$ helped to alleviate the salinity stress on anammox granules. The effect of the alleviation by the $Ca^{2+}$ augmentation was higher than that by $Mg^{2+}$. The addition of both divalent cations affected the EPS production and compositions as well as the amount of $Na^+$ entrapped in the EPS. The increased amounts of $Na^+$ associated with the EPS by the augmentation of divalent cations could contribute to the salinity stress alleviation for the anammox granules. No significant change in the microbial (anammox) community structure was observed in the anammox granules because a salt-tolerant

Ca. *Kuenenia* was predominant in the inoculum source that was used. Alleviation of the salinity stress by the augmentation of divalent cations seemed to be strongly associated with the increased EPS production and compositions rather than changes in the microbial (anammox) community compositions. Further studies on the mechanisms of the effect of salinity stress and following the alleviation by the augmentation of divalent cations on anammox activity through a long-term operation is required. Our findings are expected to improve the understanding of the granule-level mechanisms of the cation alleviation effect on the salinity stress of anammox granules and allow for the application of the anammox processes to saline wastewater treatment.

**Author Contributions:** Y.K.: Writing—original draft, conceptualization, methodology, writing, formal analysis, and investigation; J.Y.: writing—original draft, visualization, writing—reviewing and editing, and formal analysis; S.J.: investigation; J.K.: investigation and data curation; S.P.: investigation and data curation; H.B.: writing—review and editing; S.-K.R.: writing—review and editing; T.U.: writing—review and editing; S.-Q.N.: writing—review and editing; T.L.: supervision, funding acquisition, and writing—review and editing. All authors have read and agreed to the published version of the manuscript.

**Funding:** This research was supported by PNU-RENovation (2020–2021).

**Institutional Review Board Statement:** Not applicable.

**Informed Consent Statement:** Not applicable.

**Data Availability Statement:** Not applicable.

**Acknowledgments:** This research was supported by PNU-RENovation (2020–2021).

**Conflicts of Interest:** The authors declare no conflict of interest.

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
