# Peer review of "Differences in the Effects of Calcium and Magnesium Ions on the Anammox Granular Properties to Alleviate Salinity Stress"

_applsci, doi:10.3390/app12010019_

Round 1
Reviewer 1 Report
This paper aimed to investigate the effect of Ca2+ and Mg2+ augmentation on the recovery of activity of freshwater anammox granules inhibited by salinity stress. The activity, EPS contents and characteristics, morphology and microbial community of anammox were evaluated. Though the experiment methods and results are closely bound to the proposed object, whereas few conclusions are not solid. In general, the research is interesting, whereas some analysis and conclusions should be further optimized. Some detailed comments listed as follow:
- section 2.2: how long did the batch tests last? Why the authors chose the cation concentrations for the tests? “(phase 3)”was missing
- section 2.3: Are these equations from references? which reference?
- section 2.5: ‘The anammox granules were collected at the end of the A NAMMOX activity test’. It’s inappropriate to conduct the microbial community analysis, since the experimental time was too short as the authors said in line 321. That why the they got the results ‘there was no significant change in the community composition’.
4. line 159-167: more information related to the sequencing and data processing are needed.
5. line 173-174: ‘destroy the cell’ and ‘achieve their death’, please use proper words.
- line 204-208: I don’t think ‘a denser granular structure was observed in granules augmented by Ca 2+ than Mg 2+’ suggest ‘effect of divalent cations might be associated with changes in the amount and composition of EPS and/or bacterial community’.
- line 248: why the PN/PS ratio of B-EPS was considerably higher than that of S-EPS? What is behind the obtained result, and how this result relate to the alleviation mechanisms? Also, since the amount of S-EPS was rather small, can the authors conclude that ‘the S-EPS plays a more important role in maintaining anammox activity than B-EPS under saline conditions’?
- FTIR spectra results: the authors should seek some useful information beyond the results, e.g., which functional group may contribute to the alleviation of salinity inhibition.
- line 335-337: it’s inappropriate to conclude that ‘no significant change in microbial (anammox) community ….. because a salt-tolerant Ca. Kuenenia was predominant ….’.
Reviewer 2 Report
The manuscript presents well designed work that evaluates the alleviation of salinity stress on anammox bacteria using divalent cations. Although the application of this work is quite narrow, the results that rather than microbial composition, the EPS composition and production plays role in stress regulation is very interesting and can be far reaching.
Overall the introduction, methods and results sections are well written. The weakest points are some of the statements in the results that should be supported by statistical analyses. Simple ANOVA could be performed to analyze significance of the differences presented in Figures 4 and 6.
Some of the Figure bars are hard to distinguish in black and white version and different color scheme should be considered if presented in black and white.
Other minor comments:
P4L153 - there is a break in the word ANAMMOX, A on one line and NAMMOX on the other
P4L159 typo in the word using - "uusing"
Figure 1 - the x axis (salt concentration) has units mg NaCl /L, although from the text and methods it seems that it should be g NaCl/L. mg would be order of magnitude lower than reported in other studies. Additionally the exact salt concentration for IC50 should be calculated from the equation on Figure instead of just estimated. Using 5 for the follow up experiments is OK.
P5L176 instead of the word "since" it woudl be better to use "due to" or something similar.
P7L200 - the words "showed that" are used twice
P9L232 - as mentioned before especially these results and claims should be analyzed statistically. For example the PN/PS is probably not different between Mg2+ and Ca2+ statistically.
P10 mentioning possible mechanisms for these changes due to Ca2+ or Mg2+ would strengthen the results and discussion. For example in plants Ca2+ stress regulation and saline environment is extensively studied.
Fig 6 does not contain stdev, was this work performed only once or is the average presented? The Na concentrations in mg/L are confusing. I think mg/g of EPS would be more useful. Again statistical analyses will help a lot here. I am not sure the difference between Na bound in C2 and C3 is significant. C3 and C4 definitely do not look significantly different, although can not be concluded without proper analyses. It seems that less than 10% Na (mg/L compared to addition of Na) is bound in the EPS. - Not sure if these 10% or 1 % difference between C2 and C3 make any difference in the cell stress.
